# Three-Dimensional Echocardiography Assessment of Right Ventricular Volumes and Function: Technological Perspective and Clinical Application

**DOI:** 10.3390/diagnostics12040806

**Published:** 2022-03-25

**Authors:** Ashfaq Ahmad, He Li, Yanting Zhang, Juanjuan Liu, Ying Gao, Mingzhu Qian, Yixia Lin, Luyang Yi, Li Zhang, Yuman Li, Mingxing Xie

**Affiliations:** 1Department of Ultrasound Medicine, Union Hospital, Tongji Medical College, Huazhong University of Science and Technology, Wuhan 430022, China; ashfaqtajjak@hust.edu.cn (A.A.); lih0507@hust.edu.cn (H.L.); zhangytcw@163.com (Y.Z.); liujuanjuan@hust.edu.cn (J.L.); moai1194563986@126.com (Y.G.); qianmingzhu95@hust.edu.cn (M.Q.); linyixia@hust.edu.cn (Y.L.); yiluyang@hust.edu.cn (L.Y.); zli429@hust.edu.cn (L.Z.); 2Clinical Research Center for Medical Imaging in Hubei Province, Wuhan 430022, China; 3Hubei Province Key Laboratory of Molecular Imaging, Wuhan 430022, China; 4Shenzhen Huazhong University of Science and Technology Research Institute, Shenzhen 518057, China; 5Tongji Medical College and Wuhan National Laboratory for Optoelectronics, Huazhong University of Science and Technology, Wuhan 430022, China

**Keywords:** right ventricular volumes, right ventricular function, three-dimensional echocardiography, ejection fraction, cardiac magnetic resonance

## Abstract

Right ventricular (RV) function has important prognostic value in a variety of cardiovascular diseases. Due to complex anatomy and mode of contractility, conventional two-dimensional echocardiography does not provide sufficient and accurate RV function assessment. Currently, three-dimensional echocardiography (3DE) allows for an excellent and reproducible assessment of RV function owing to overcoming these limitations of traditional echocardiography. This review focused on 3DE and discussed the following points: (i) acquisition of RV dataset for 3DE images, (ii) reliability, feasibility, and reproducibility of RV volumes and function measured by 3DE with different modalities, (iii) the clinical application of 3DE for RV function quantification.

## 1. Introduction

Right ventricle is formerly thought to be a non-essential cardiac chamber that offers little contribution to overall cardiac function. Nowadays, more and more investigators focus on right ventricle and demonstrate that right ventricular (RV) function is essential in the management of patients with a variety of cardiovascular diseases. Indeed, published data have shown that RV function is independently related to poor clinical outcomes in individuals with various cardiopulmonary pathologies [1,2]. Current research reveals that RV function is an important independent predictor of morbidity and mortality in patients with heart failure (HF), congenital heart disease (CHD), pulmonary hypertension (PH), and coronary artery disease (CAD) [3]. Another recent study also demonstrates an undeniable link between RV hypertrophy and the risk of HF and sudden cardiac death in a multiethnic population free of known cardiovascular disease [4]. Cardiac magnetic resonance (CMR) imaging is still the gold standard for assessing RV volumes and function [5,6,7], but it is not feasible for patients who have common contraindications to implanted cardiac medical devices or claustrophobia. Cardiac computed tomography (CT) also provides accurate and reproducible evaluations for RV volumes and function in comparison with CMR [8]. However, it is radioactive. Echocardiography is the first-line tool for assessing the right ventricle because of its availability, simplicity and reduced cost. Two-dimensional echocardiogram (2DE) is commonly utilized to evaluate RV function in clinical practice. However, 2DE parameters such as tricuspid annular plane systolic excursion (TAPSE), peak systolic velocity (s’), and fractional area change (FAC) are angle-dependent and relatively reproducible, and provide less accurate quantification of RV function [9,10,11].

Due to the complex structure of right ventricle, three-dimensional echocardiogram (3DE) provides a more accurate and reliable assessment of RV function than 2DE owing to the avoidance of geometric assumptions and the foreshortened images [12]. With the emergence of novel echocardiographic technologies, particularly 3DE, reliable measurement of the right ventricle has become possible [13,14,15]. Many studies have established the accuracy of 3DE for RV measurements in comparison with CMR [16,17,18,19] and have also confirmed its incremental value over 2DE [20,21,22]. Firstly in a broad number of healthy people, reference values for RV volumes and ejection fraction were obtained using real-time 3DE [23]. Despite these aforementioned advantages, widespread use of 3DE for RV measurements has not been established in daily clinical practice because it is relatively time-consuming and requires great expertise to obtain an accurate 3DE image for RV measurements. Different 3DE semi-automatic and fully automated RV quantification software, based on machine learning algorithms (MLA) as well as three-dimensional speckle tracking echocardiography (3D-STE), has recently been developed. The goal of this review was to give a comprehensive overview of the 3DE and 3D-STE for evaluating RV volumes and function, with an emphasis on the clinical application.

## 2. Right Ventricular Anatomy

Right ventricle is the most anterior cardiac chamber, bordered anteriorly by the sternum. Anatomically, right ventricle consists of three parts: RV inflow component (i), the trabeculated muscular apex (ii), and the outflow component (iii) [24,25]. Right ventricle has a triangular shape in the coronal plane and a crescent shape in the transverse plane [26]. The RV inflow and trabeculated muscular apical components are utilized to measure regional function using speckle tracking echocardiography (STE) [27,28]. However, there is no discrete boundary between adjacent portions. Moreover, the right ventricle consists of anterior, inferior, and lateral walls, each of which is further divided into basal, mid, and apical parts. The inlet component of the right ventricle consists of the tricuspid valve complex, which comprises the tricuspid valve, chordae tendineae, and the three papillary muscles that originate in the ventricular wall and are attached to the anterior, posterior and septal leaflets. Each papillary muscle cord is attached to two adjacent leaflets. The largest papillary muscle is the anterior papillary muscle, while the septal papillary muscle is the smallest one. The septal papillary muscle is located where the crista supraventricularis meets the posterior arm of the septomarginal trabeculation and provides attachment to the cord to the septal and posterior leaflets.

Three thick intracavitary muscles, including the septomarginal trabeculation, crista supraventricularis, and moderator band, are present in the trabeculated muscular apical component of the right ventricle. Both the septomarginal trabeculation and septal band are attached to the septum in a Y-shaped manner. The outlet component of the right ventricle is separated from the inlet component by crista supraventricularis. The outlet component is a smooth funnel shaped tract called the sub-pulmonary infundibulum. The sub-pulmonary infundibulum part extends from the crista supraventricularis to the pulmonic valve.

## 3. Right Ventricular Mode of Contractility

The right ventricle has the following three wall motions: (i) the free wall of the right ventricle moves inward (ii) long-axis shorting occurs due to the action of deep muscle longitudinal fibers, which is aligned longitudinally from base to apex. (iii) traction to the free wall occurs at the point of attachment secondary to the left ventricle. RV contraction is mainly dependent on longitudinal contraction rather than inward motion [29]. RV contraction occurs in a sequential manner, starting with the contraction of the inlet and trabeculated myocardium and ending with the contraction of the infundibulum [30]. Left ventricular (LV) contraction is thought to account for 20–40% of RV volume [31]. Heart rhythm, RV systolic synchronization, atrioventricular synchrony, and ventricular interdependence have an important effect on the global contractility of the right ventricle. Ventricular–ventricular interdependence is mostly mediated by the interventricular septum [32].

## 4. Acquisition of RV Dataset for 3D Images

In an ideal scenario, a 3D dataset with a time resolution of more than 20 volumes per second would contain the whole right ventricle. Inadequate covering is typically caused by the RVOT and the RV’s anterior wall [33]. The RV-focused apical four-chamber view is advised to acquire a RV 3D image [12,16]. The RV-focused apical four-chamber image is obtained by retaining the LV apex at the center of the scan line and ensuring the maximum basal diameter of the right ventricle with a more lateral transducer position than the one used for the conventional apical four-chamber view [34]. With a slight rotation of the transducer, a more cranial intercostal space can occasionally aid in visualizing both the tricuspid and pulmonic valves.

Different ultrasound equipment’s uses for the acquisition of RV data set for 3D images nowadays including TomTec-Arena (TomTec imaging systems, Unterschleissheim, Germany) [35], ARTIDA ultrasonography system (Toshiba Medical Systems, Tochigi, Japan) [27], Philips Healthcare, Andover MA [16,36,37,38], GE Vingmed Ultrasound, Horten, Norway [19,39], Toshiba Medical Systems [28], and Philips Medical Systems, Best, the Netherland [40,41] are widely used.

The 3D data set should be stored and utilized for further research offline. The 3D data set should be subjectively evaluated to ensure that the RV endocardial border is visible in all three views (apical four-chamber, coronal and basal short axis). The signal-to-noise ratio will be used to grade the 3D image quality, which should range from 1 to 4 (poor to excellent) [19]. If the echocardiographic dropout in coronal view covers more than half of the RV free wall, the image quality is rated poor [42]. Commercially available software for 3D quantification including 4D-RV function version 2.0 [16,19,35,37], 3D viewer software (QLAB, Philips Healthcare) [36], 4-D RV function version 4.0 [41], Ventripoint Medical System (VMS) version 1.2.6804.1278 [40], 3-D speckle tracking software (4D RV analysis ver.2.0 (TomTec imaging system GmbH) [38], semi-automated vendor-independent 3DE RV quantification software [43] and EchoPac v201 [44] are commonly utilizing in daily practice. the software automatically detects the RV endocardial border using artificial intelligence. The newly simplified on board (OB) 3DE software not only allows semi-automated or fully automated analysis for RV volumes and function but also derivate several parameters such as TAPSE, RV-FAC and strain analysis [45]. A study by Tamborini, G. et al. demonstrate that the newly simplified OB 3DE RV reconstruction software is feasible and being implemented on the echocardiographic machine, which not only reduces the time necessary to obtain 3DE volumes and function in daily practice but also rapid on board analysis and avoids of RV data set loading and calculation on the off-line system [45]. No significant difference was observed between OB 3DE and conventional 2DE parameters (TAPSE, FAC) analysis. Moreover single-beat full-volume 3DE is a feasible technique for RV size and function quantification [46]. The process of 3DE RV data set analysis is shown in Figure 1. Recently, the 3D fully automated software using artificial intelligence approaches, including MLA, has enabled automated detection of the RV endocardial border from the 3DE data set (3D auto RV on QLAB, Philips Health Care, Andover, MA, USA). This software enabled the accurate measurements of RV volumes and function [47]. The completely automated ML evaluation of the right ventricle was accurate in one-third of patients, which was lower than the LV analysis due to poor image quality in the anterior wall and RVOT, which influenced accurate measures [36]. As a result, 3DE is recommended for the right ventricle quantification in clinical setting.

Although the RV-focused apical four-chamber view is recommended to measure RV volumes and EF, some investigators quantify RVEF using the apical four-chamber view (A4CV). The feasibility of RVEF measurements on 3D imaging from the RV-focused apical four-chamber view and A4CV was found 92% for each of them with a good correlation (r = 0.83) and small bias (0.3%) between RVEF from the RV-focused apical four-chamber view and A4CV [48]. However, De Potter T et al. revealed that feasibility of imaging acquisition was higher for the RV-focused apical four-chamber view (80.0%) then the standard A4CV (16.7%) [39]. The similar trend was also observed by the study of Medvedofsky, D. et al., which showed that the RV-focused apical four-chamber view was feasible in most of the patients [36].

## 5. Reliability, Feasibility, and Reproducibility of RV Volumes and Function with Three-Dimensional Echocardiography

CMR remains the gold standard imaging modality of RV volumes and function [43], hence, CMR measurements have been used to test the reliability of RV quantification software [13,19,20,28,36,38,39,40,41,43,46,48,49,50,51,52]. Age-, body size- and gender-related specific reference values for RV volumes and function by 3DE in a multicenter echocardiographic study in healthy volunteers are presented by Maffessanti, F. et al. [53] as well as normal value for RV function in comparison with CMR [54]. The studies suggests that 3DE method for the assessment of RV volumes and function are feasible and relatively simple and not time consuming in offline as well as on board analysis [45,55]. The comparative study of the 3DE with CMR are summarized in Table 1. Different 3DE software such as 4D RV, 4D RV function, STA-3DE auto, fully auto 3DE RV, 3DE STE system, RT 3DE software etc. were used in these observations. Almost all of the participants had various cardiac disorders, primarily dilated cardiomyopathy, coronary artery disease, valvular heart disease, and so on. The 3DE software’s feasibility was evaluated as a percentage of available case numbers for 3DE analysis out of an initially enrolled case number. The feasibility ranged from 80% to 96%. The most common reason for the subject’s elimination was due to exceedingly low imaging quality in the RVOT and anterior wall.

Consistently, the 3DE RV software shows a strong correlation with CMR in RV volumes and function. The correlation coefficients are almost above 0.7 for RV volumes (RVEDV and RVESV). The 3DE underestimates the RV volumes [13,16,19,28,36,37,38,39,41,43,46,48,49,52,56], except in the case of Muraro, D. et al. study for which RVESV was overestimated [19]. The mean differences were as followed: −2.3 to −53 mL for RVEDV and −0.3 to −23.6 mL for RVESV. The lowest bias for RVEDV and RVESV was found in the study of Medvedofsky, D. et al. with novel 3DE [16].

The correlation coefficients of 3DE-derived RVEF with CMR values in most of the studies exceed 0.75, and the mean difference for RVEF between 3DE and CMR is discordant. Some studies showed that 3DE slightly underestimates the RVEF against CMR [16,28,36,39,41,43,46,49,56]. The mean bias for RVEF ranged from −0.3% to −17.0%. While other observations reported the overestimation of RVEF against CMR and the bias ranged from 0.4% to 17.4% [19,37,38,52]. While no significant difference was found in Laser, K.T. et al., Otani, K. et al. and Leibundgut, G. et al. study [13,40,43]. The lowest bias was found in the study of Laser, K.T. et al. with the RT-3DE [40]. Intra-and interobserver variability for 3DE software for RV volumes and RVEF are highly reliable as reflected the variability from 0 to 0.99% reported by Otani, K. et al., Ahmad, A. et al., Knight, D.S. et al. and Laser, K.T. et al. [40,43,46,52]. Test–retest variability is found in the range of 4.3–7.8% for novel 3DE auto RV with a good ICC above 0.8. For the manual edit method, the variability ranges from 3.3% to 8.7%. [43,52]. As a result, the novel 3DE software is highly reproducible.

## 6. Impact of RV Function and Frame Rate on 3DE Software

The degree of RV function greatly impacts 3DE measurements. Ahmad et al. showed that the biases for RV volume and EF were higher in patients with lower RVEF [52]. Similarly, Tsang et al. found that the bias and limit of agreement (LOA) for LV volumes using 3DE auto LV quantification software were larger in subjects with lower LV ejection fraction [57]. In addition, the extent of underestimation of RV volumes was greater in patients with RV dilatation than those with normal right ventricle [18]. As dilated right ventricle and numerous RV trabeculae are seen in subjects with severe decreased RV function.

Recently, a fully automated 3DE quantification software has been developed and performs better with a high frame rate than with a low frame rate regardless of manual editing. Ahmad, A. et al. found that the mean bias and LOA for RV volumes were higher in subjects with low frame rate [52]. However, RVEF was unaffected by frame rate [52,58]. Similarly, Tsang et al. also showed that the mean bias and LOA for LV volumes were higher in participants with low frame rate [57].

## 7. Clinical Application

### 7.1. Prognostic Value of RV Function in Patients with Various Cardiovascular Diseases

RVEF evaluated by 3DE has been considered to be an independent predictor of adverse clinical events in a large number of patients with various cardiovascular diseases [20,36]. Nagata, Y. et al. investigated 446 patients with various cardiovascular diseases and 88 major outcomes occurred during a median of 4.1 year follow up. Univariable Cox proportional analysis revealed that 3D-RVEF was associated with both cardiac death and major cardiac outcome [20]. Namisaki, H. et al. found that 21 patients out of 174 patients experienced primary end point with a median follow-up period of 12.5 months. They demonstrated that RVEF using fully automated analysis has significant association with cardiac events (RV focus view: hazard ratio [HR], 0.90 [*p* = 0.009, *n* = 44]; A4CV: HR, 0.90 [*p* = 0.009, *n* = 68]) [48]. Another study by Muraru, D. et al. also revealed the independent prognostic value of RVEF assessed by 3DE in patients with cardiac disease, and showed that RVEF could help to stratify the risk of cardiac death and major adverse cardiac events [58].

### 7.2. Pulmonary Hypertension

RV dysfunction is independently associated with survival in PH and has been regarded as an independent predictor of adverse clinical outcomes in patients with PH [21]. Interestingly, 3DE is an ultimate alternative with respect to CMR due to the complex RV anatomy. The clinical applications of 3D-STE are shown in Table 2. In a study of 96 pediatric PH patients, Jone, P.N. et al. demonstrated that PH patients displayed higher RV volumes, and lower RVEF, free wall and septal RV longitudinal strain, TAPSE and FAC than those in normal controls. Moreover, 3D-RVEF, free RV longitudinal strain and FAC were independently associated with adverse clinical outcomes in patients with PH (3D-RVEF: HR 0.1, 95% CI 0.03–0.27, *p* < 0.001; free RV longitudinal strain: HR 0.17, 95% CI 0.07–0.45, *p* < 0.001; FAC: HR 0.08, 95% CI 0.03–0.22, *p* < 0.001) [59]. In a prospective cohort of 104 subjects with PH along with 34 controls, 3D RV semi-automated (Tom tec 4D RV-Function 2.0) were used. Over 6.7-month follow-up, 16 patients died. Cox proportional hazards analysis showed that 3D-RVEF and global RV area strain (AS) were independent predictors of clinical events in PH. Kaplan–Meier analysis showed that patients with 3D-RVEF less than 38% had significantly event-free survival than those with greater than 38% (*p* = 0.0007). Global RV AS > −18% was the most powerful RV function parameter for identifying patients with increased risk of death [60]. Murata, M. et al. shows the strongest correlation of 3D-RVEF with hemodynamics followed by 6-min walk. Characteristic analysis of association with cardiac events revealed a greater AUC for 3D-RVEF than that for mean pulmonary arterial pressure (0.78 vs. 0.74) [21]. In a recent study by Vitarelli, A. et al., they have studied 73 adult patients with chronic PH of a different etiology using echocardiography and cardiac catheterization. Standard 2D measurements and 3D RV volumes and global and regional ejection fraction (3D-RVEF) were determined. The 3D-RVEF was lower in patients with precapillary PH (*p* < 0.0001) and postcapillary (*p* = 0.004). Furthermore, 3D-RVEF (HR: 5.3, 95% CI 2.85 to 9.89, *p* = 0.002) was independent predictor of mortality. Receiver operating characteristic curve showed that threshold offering an adequate compromise between sensitivity and specificity for detecting hemodynamic signs for RV failure were 39% for 3D-RVEF (AUC 0.89) and −17% for 3D longitudinal strain of RV free wall (AUC 0.88). Hence, they showed that 3D-STE parameters indicated global and regional RV dysfunction that were associated with RV failure hemodynamics better than conventional echo indices in chronic PH [61]. Additionally, in a prospective study of 66 with acute episode and after 6 months follow-up of acute pulmonary embolism patients by Vitarelli, A. et al., they found that 3D-RVEF was lower in patients with pulmonary embolism than controls. Receiver operating characteristic curve analysis showed that 3D-RVEF was the most powerful predictor of adverse events. Multivariate analysis demonstrated that RV systolic pressure (*p* = 0.007), mid free wall of RV longitudinal strain (*p* = 0.002) and 3D-RVEF (*p* = 0.001) were independently associated with adverse outcomes [62]. Similarly, the use of RT-3DE to determine RV regional systolic performance could help in the noninvasive assessment of PH severity [63]. The 3D-STE quantification of right ventricle revealed that RVs area strain most closely associated with RV function and provided valuable prognostic information regarding clinical outcomes independent of other variables [64].

### 7.3. Heart Failure

Half of HF patients have preserved ejection fraction, which has been identified as a key cause of cardiovascular mortality [65]. Previous studies show that RV dysfunction is a strong predictor of morbidity and mortality in individuals with heart failure with preserved ejection fraction (HFpEF) [66,67,68]. Hence, accurate analysis of RV function is highly desirable and needed for treatment and management. Conventional RV functional parameters, such as RVFAC, S’ and TAPSE, predict the prognosis of HF [69], but all have its own limitations [9,12]. Two-dimensional (2D) STE has been considered to be a sensitive and reliable quantitative tool for RV function assessment [70,71,72]. Furthermore, 2D-STE derived right ventricular longitudinal strain of free wall (RVFWLS) provides incremental prognostic information in patients with PH, HF with reduced EF (HFrEF) and corona virus [1,51,73]. However, 2D-STE has some limitations including foreshortened view, geometric modeling and out of plane motion of the spackles. The 3D-STE allows a more accurate evaluation of RV function owing to overcoming the aforementioned limitations of 2D-STE [19,27]. Recently, a total of 93 consecutive patients with HFpEF was investigated using 3D-STE by Meng, Y. et al. [74]. With a median follow-up time of 17 (11–36) months, 39 (48%) of patients reached the end point. This study revealed that 2D- and 3D-RVFWLS and RVEF were independently associated with poor clinical outcome in patients with HFpEF. Moreover, a multivariate Cox hazard model revealed that 3D-STE have a similar prognostic value to 2D-STE [74]. In another cohort study of 124 subjects with end stage HF for predicting right ventricular myocardial fibrosis (RV MF) against the histological confirmation of MF, Tian, F. et al. demonstrated that 3D-RVFWLS was a strong predictor of RV MF compared with that of 2D-STE and conventional RV function parameters, indicating that 3D-STE might be an accurate tool to detect MF in patients with end stage HF [75]. In addition, a study involving 59 patients with aortic valve disease and 48 control subjects, demonstrated that 3D-STE are useful indices of HF in early stage caused by aortic valve disease [76]. A study by Lu, K.J. et al. also demonstrated that RV global longitudinal strain (GLS) measured by 3D-STE best predicted the presence of RV dysfunction as defined as RVEF < 48% on CMR (hazard ratio = 7.0 [1.5–31.7], *p* < 0.01). Receiver operator characteristic analysis revealed that RV GLS of −20% was the most specific and sensitive predictor of RV dysfunction (AUC 0.8 [0.57–1.0]. *p* < 0.02) [15].

### 7.4. Congenital Heart Disease

Further, 3DE has valuable clinical application in patients with CHD and is promising in condition especially complex right ventricle or functional single ventricle and live assumptions in interventions [77]. Multidetector computed tomographic and CMR is still consider gold standard in adult CHD but the former limited due to radiation exposure and later common contraindicated cardiac magnetic devices. Additionally, 3DE has been widely used for RV assessment. RV dysfunction predicts poor outcomes in patients with CHD. Regarding atrial septum defect (ASD), one of the common congenital heart diseases in adults as well as in children, RV function assessed by 3DE provide a valuable prognostic information in such patients. Vitarelli A et al. sought to evaluate RV function using 3DE and myocardial strain imaging in 39 adult patients with ASD before and after 6 months follow-up. Apical strain and strain rate were found to be independent predictors of NYHA functional class in multivariate analysis. When compared to 2D-Doppler indices, ROC analysis revealed that 3D-RVEF and apical strain were more sensitive predictors of adverse outcomes after defect closure [78]. Additionally, a study for assessing RV global and regional EF using real time (RT) 3DE within 24 h before and after percutaneous closure in 81 patients with ASD, revealed that RV global and regional EF was impaired in open and closed ASD. RT-3DE derived parameters were negatively correlated with RV afterload [79].

Patients with hypoplastic left heart syndrome after Fontan palliation underwent 3D-STE examination to measure RVEDVi, RVEF, and RV GLS. Volume measurements were compared between 3D-STE and 3DE, and strain parameters were compared with 3D-and 2D-STE. Strong correlation was observed between RVEDVi and RVEF by 3D-STE in comparison with 3DE (r = 0.93 and 0.87, respectively), while RVGLS shown moderate correlation between 3D- and 2D-STE [80]. Additionally, a study by Ishizu, T. et al. using isochrone activation imaging (AI) system with 3D-STE in Wolf–Parkinson–White (WPS) syndrome reveals that isochrone AI system with 3D-STE may be a promising noninvasive tool for the assessment of cardiac synchronized activation in normal heart and detect abnormal breakthrough of mechanical activation from both atrioventricular annuli in WPW syndrome [81].

The accurate evaluation of RV function has important prognostic significances in patients with repaired tetralogy of Fallot (TOF). The 3DE provide valuable information in such patients. In a study of 41 subjects with TOF, the investigators aimed to evaluate RV function pre- and post-operative using real-time 3DE. RV volumes and function was assessed before surgery, seven days and three months after surgery. Correlation between preoperative Nakata index and postoperative RV function was analyzed. The postoperative 7-day and 3-month RVEDV and RVESV were not different (*p* > 0.05) when compared to the RVEDV and RVESV before surgery. In contrast, postoperative RVEF decreased compared with preoperative values (*p* < 0.05). Therefore, RVEF assessed by 3DE provides clinical significance in determining postoperative efficacy [82]. Similarly, a study including 41 patients with repaired TOF and 20 control subjects revealed that patients with repaired TOF displayed characteristic RV remodeling measured by 3DE. The largest volume was observed at the apical region as compared to control patients, but RVEF at the inlet and outlet was significantly decreased [83].

### 7.5. Valvular Heart Disease

Echocardiography has been used to evaluate and diagnose patients with valvular heart disease. With the development of 3DE, it is used as one of the most promising methods for the diagnoses of valvular heart disease. Among the four-heart valve, 3DE is widely used for evaluating mitral valve disease. Many studies have reported the usefulness of 3DE in assessing the mitral valve disease especially in patients with mitral regurgitation [84,85,86,87,88,89,90,91]. In a prospective multimodality imaging study of 90 ischemic mitral regurgitation patients assessed by stress CMR and 3DE, Jiwon Kim et al. concluded that RV dysfunction was associated with potentially reversible process, and strongly impacted by volumetric loading condition in patients with ischemic mitral regurgitation [92]. Another observational study of 45 patients undergoing percutaneous mitral valve replacement (PMVR) showed that the post-PMVR RVEF (OR 1.15: 95% CI 1.02–1.29; *p* = 0.02) and the change in RVEF (OR 1.13: 95% CI 1.02–1.25; *p* = 0.02) were significant predictors of improved clinical outcome at 6-month follow up. They concluded that RV function may be an key noninvasive predictive parameter, demonstrating that PMVR treatment of severe mitral regurgitation may have a therapeutic advantage [93]. Additionally, in a study of 42 patients with mitral valve repair, RVEF was calculated using 2D STE and 3DE before and 6 months after mitral valve repair. The researchers demonstrated that 3D-RVEF was preserved after valvular surgery. However, RV longitudinal strain was decreased [94]. Moreover, in the assessment of biventricular function in patients with mitral regurgitation after MitraClip, 3D-STE showed overall biventricular strain improvement after clip implantation and lower post procedural LV strain in patients with worse preexisting RV function. These results could help in guiding mitral regurgitation management. In the case of severe RV impairment, other therapy may be suggested, or the procedure may be planned in advance if RV dysfunction worsens [95]. Similarly, 3D-STE may provide sufficient information regarding left and right ventricular function in patients with mitral stenosis, as shown by Seckin Gobut, O. et al. demonstrating that subclinical LV and RV systolic dysfunction were present in mild-moderate mitral stenosis patients [44]. With respect to the aortic valve and considering its 3D structure, the aortic valve may prove to be one of the most important application of 3DE [96,97,98]. In comparison with 2DE, 3DE increase accurate identification of abnormal valve leaflets especially bicuspid and quadricuspid aortic valve, and may be useful in the assessment of aortic valve masses such as Lambl’s excrescences and papillary fibroelastoma [99]. Previous studies revealed the prognostic utility of RV functional parameters in aortic stenosis [100,101]. RVEF assessed by 3DE has important prognostic value in patients with aortic stenosis. In a retrospective study of 392 patients with asymptomatic aortic stenosis with a median follow-up of 27 months, multivariate analysis revealed that RVEF was significantly associated with cardiac events [102].

### 7.6. Myocardial Infarction

Left ventricular function is a well-established predictor of ventricular arrhythmia in patients with myocardial infarction (MI) but little is about RV function. The most common complication after acute myocardial infarction is ventricular arrhythmias, HF, or even sudden cardiac death. Among patients with myocardial infarction, there is a strong relationship between the degree of HF and mortality [103]. The 2D RV strain is shown to be significantly and independently related to ventricular arrythmias and sudden cardiac death in patients with acute MI [104]. A study by Zamfir, D. et al. showed that RVFWLS (OR: 1.04; 95% CI: 0.21–5.08) and 3D-RVEF (OR: 0.83; 95% CI: 0.20–3.43) were predictive for in hospital major adverse cardiovascular events regardless of the culprit coronary artery [105].

In patients with suspected acute MI especially inferior wall MI, RV function plays an important role in diagnosis as well as in management. In such situation, 3DE plays a vital role in the evaluation of the RV performance [106]. In a study of 85 patients with acute MI complicated with RV myocardial infarction admitted for percutaneous coronary intervention and similar number of patients with isolated inferior wall MI served as control. RV function were assessed by 3DE in all patients before percutaneous coronary intervention. Their findings showed that RVEF was lower in RV myocardial infarction patients than in controls (41.7 ± 6.03 vs. 52.7 ± 2.3%, respectively). The cutoff value of RVEF was <51% for diagnosis of RV myocardial infarction with the sensitivity of 91% and the specificity of 80%, and may be a useful diagnostic index in such patients [107].

### 7.7. Post Cardiac Surgery

The importance of the right ventricle in determining exercise capacity and the predictive value of RV function in HF, as well as the success of cardiac surgery, has been fully established [108,109]. RVEF does not change after surgery when RV global systolic function is evaluated by RT-3DE [110]. Another study by Maffessanti, F. et al. demonstrated that there is no change in RV function using 3DE after mitral valve repair [94]. Moreover, biventricular function plays a vital role in the prognosis of heart transplantation. Biventricular mechanical function assessed with 3DE, were decreased in clinically well pediatric heart transplantation patients [111]. A comparative study of RV function between surgical replacement of aortic valve (SAVR) and transcatheter aortic valve implantation (TAVI) suggest that TAPSE was postoperatively markedly decreased in patients undergoing SAVR but no change was observed in patients undergoing TAVI, however, RV function assessed by 3DE remain unchanged in both (SAVR and TAVI) patient groups [112]. In a study by Cronin, B. et al., patients who underwent elective pulmonary thromboendarterectomy (PTE) surgery were assessed by 3DE pre-and-post PTE surgery. The 3DE results showed that there was no statistically significant difference in RVEF or RVFAC after PTE compared with pre-PTE results [113].

**Table 2 diagnostics-12-00806-t002:** Assessment of RV function in patients using 3D-STE.

References	Disease	Sample Size (*n*)	Age (Years)	Men, *n* (%)	RVEF (%)	3D-STE Parameters	Main Findings
Lu, K.J. et al. [15]	HF	60	45 ± 10	60%	53 ± 8	RVGLS	RV GLS best predicted the presence of RV dysfunction,
Seckin Gobut, O. et al. [44]	MS	20 ^C^ 20 † 20 ^#^	46.9 ± 11.6 46.9 ± 10.4 49.6 ± 11.8	13 (65%) 13 (65%) 13 (65%)	–	RV-FWLS	RV deformation indices showed significant decrease in correlation with the severity of the mitral stenosis
Jone, P.N. et al. [59]	PHT	96	8.1 ± 5.2	53 (55%) *	46 ± 5	RV LS free wall and septum	PH patients have impaired RV function compared with normal children. 3D RV EF, volumes, FAC, and free wall RV strain serve as outcome predictors for PH patients.
Moceri, P. et al. [60]	PHT	104	65.9 [62.0–68.8]	58 (55.8%) *	35.6 ± 9.7	RVGLS, RVCS, RVAS	RV strain patterns gradually worsen in PH patients and provide independent prognostic information. This technique could help better stratify the risk in PH patients.
Vitarelli, A. et al. [61]	PHT	73	53 ± 11	44%	35.5 ± 7.6	RVGLS, RV-GFW, RV-FWAS	In PH patients, the quantitative assessment of global and regional RV function by 3D and STE provides useful hemodynamic and prognostic information.
Vitarelli, A. et al. [62]	PE	66 66 ^c^	53 ± 11	32	37 ± 8	MFW RVLS	Decreases in MFW RVLS and 3D RVEF may persist during short-term and long-term follow-up and correlate with unfavorable outcomes
Smith, B.C.F. et al. [64]	PHT	97	60.6 ± 15.3	34 (35)	31.4 ± 9.6	RVGLS, RVCS, RVAS	AS best correlated with RVEF and provides prognostic information independent of other variables.
Meng, Y. et al. [74]	HFpEF	81 (*n* = 42) ^a^ (*n* = 39) ^b^	61 ± 12 ^a^ 63 ± 13 ^b^	27 (64%) ^a^ 26 (67%) ^b^	47 ± 4 ^a^ 44 ± 5 ^b^	3D-RVFWLS	3D-STE parameters were powerful predictors of poor outcomes, providing similar predictive values as 2D-STE indices in patients with HFpEF.
Tian, F. et al. [75]	HF	105	44 ± 16	17 (16%) *	26.89 ± 8.09	3D-RVFWLS	3D-RVFWLS could be a promising noninvasive parameter in identifying severe MF in patients with end-stage HF
Sato, T. et al. [80]	CHD	64	10.6 (2.4–18.4)	28 (43.8%) *	51.2 (22.9–64.2)	GLS, GPS, GCS	Analysis of a single 3D-STE clip of the cardiac cycle provides useful information regarding both volume and the functional status of HLHS, which can be useful during longitudinal follow-up as outpatients.
Ishizu, T. et al. [81]	WPW	38	42 ± 21	22 (57%)	–	3DSTE (AI)	Isochrone AI with 3D-STE may be a promising noninvasive imaging tool to assess cardiac synchronized activation in normal hearts and detect abnormal breakthrough of mechanical activation from both atrioventricular annuli in Wolff–Parkinson–White syndrome.
Cui, C. et al. [82]	TOF	20 22 ^c^	24.7 ± 8.6 20.6 ± 7.0 ^c^	12 11	28.1 ± 64.4 31.9 ± 63.8 ^c^	RV-GAS	progressive RV dysfunction in patients with repaired TOF.
Vitarelli, A. et al. [95]	MVD	32	79.4 ± 5.5	18 (56.2%)	53.6 ± 7.2 ^c^ 42.5 ± 7.6 ^β^ 52.2 ± 7.9 ^Ω^	GLS, FWLS	3D-STE showed overall biventricular strain improvement after clip implantation and lower post procedural LV strain in patients with worse preexisting RV function.

a: without endpoint, b: with endpoint, c: control, β: Baseline, Ω: 6-months follow-up, †: Mild MS, #: Moderate MS, *: Female.

## 8. Conclusions

RV function is an important independent predictor of morbidity and mortality in patients with a variety of cardiovascular diseases, including HF, CHD, PH, and CAD. RV function assessed by 3DE provided valuable diagnostic and prognostic information in these patients which allows us to better understand and evaluate complicated RV pathology. With the technological improvements in spatial and temporal resolutions, and image quality of 3DE, it may play a more vital role in the accurate and comprehensive assessment of the right ventricle in daily clinical practice.

## Figures and Tables

**Figure 1 diagnostics-12-00806-f001:**
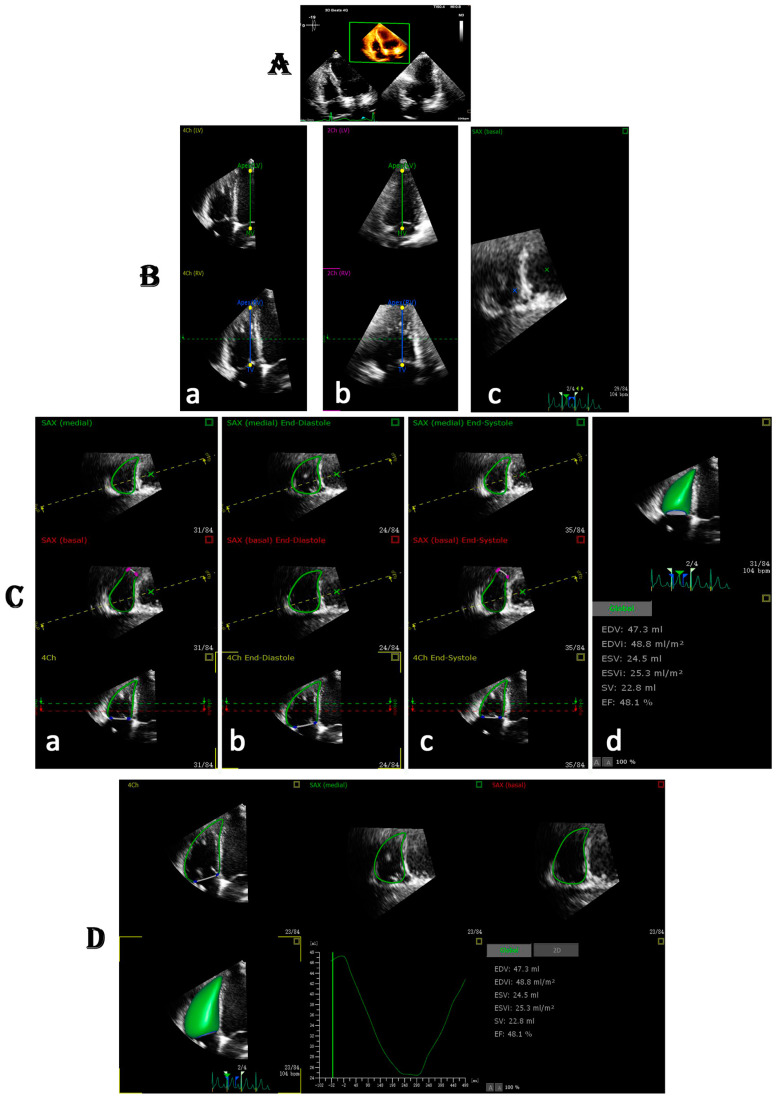
Shows the process of Auto RV analysis. (**A**) retrieving the RV focus three-dimensional echocardiography (3DE) data set aiming for RV analysis, (**B**) the software automatically adjusts the five landmarks: 4C LV and RV (a); 2C LV and RV (b) and basal SAX (c), (**C**) and the software automatically determine the RV border at 4C view (a); at end-diastole (b); end-systole (c) and retrieve the global analysis (d). (**D**) and provide the results within 15 s.

**Table 1 diagnostics-12-00806-t001:** Comparison of 3DE RV quantification with cardiac magnetic resonance.

References	Sample (*n*)	3DE Tool	RVEDV					RVESV					RVEF				
			CMR RV-EDV	3DE RV-EDV	r	Bias LOA	*p* Value	CMR RV-ESV	3DE RV-ESV	r	Bias LOA	*p* Value	CMR RV-EF	3DE RV-EF	r	Bias LOA	*p* Value
Leibundgut, G. et al. [13]	100	RT 3DE	134.2 ± 39.2	124.0 ± 34.4	0.84	10.2 (−31.3–51.7)	<0.001	69.7 ± 25.5	65.2 ± 23.5	0.83	4.5 (−23.8–32.9)	<0.02	48.2 ± 10.8%	47.8 ± 8.5%	0.72	0.4 (−14.2–15.1)	0.57
Lu, K.J. et al. [15]	60	RT3DE	188 ± 69	171 ± 48	0.74	23 (−65–111)	<0.001	91 ± 47	85 ± 36	0.85	11 (−33–55)	<0.001	53 ± 8	53 ± 8	0.56	−0.1 (−14.1–14.1)	>0.05
Medvedofsky, D. et al. [16]	147	novel 3DE	183 ± 66	172 ± 61	0.95	–11± 20	0.17	102 ± 57	101 ± 55	0.96	−0.3 ± 15.3	0.96	47 ± 13	44 ± 13	0.83	−3.3 ± 7.6	–
Muraru, D. et al. [19]	47	4D RV fn STA-3DE (auto)	–	–	0.89	–27 ± 54	<0.0001	–	–	0.82	10 ± 40	<0.0001	–	–	0.36	−17.0 ± 19.0	0.021
		Manual	–	–	0.92	–15 ± 45	<0.0001	–	–	0.93	−4 ± 28	<0.0001	–	–	0.86	1.4 ± 9.7%	<0.0001
Ishizu, T. et al. [28]	75	3D STE System	127 ± 69	118 ± 71	0.88	−9.1 (−56–38.7)	<0.001	84 ± 54	81 ± 55	0.88	−1.7 (−39.6–33.3)	<0.001	35 ± 12	32 ± 11	0.71	−2.3 (−14.7–9.9)	<0.001
Medvedofsky, D. et al. [36]	30	4D-RV Contrast	192 ± 56	176 ± 46	0.92	−16 ± 23	0.00	103 ± 44	92 ± 36	0.92	−10 ± 16	0.00	47.7 ± 6.10	48.4 ± 11	0.87	0.7 ± 5.5	0.25
		without Contrast	192 ± 56	156 ± 49	0.90	−36 ± 25	0.00	103 ± 44	79 ± 35	0.92	−23 ± 18	0.00	47.7 ± 6.10	50.5 ± 11	0.70	2.7 ± 8.1	0.25
Genovase, D. et al. [37]	56	MLA 3DE	176.6 ± 50.3	151.0 ± 50.0	0.91	−25.6 (−66.9–15.6)	<0.001	88.0 ± 38.5	80.5 ± 37.4	0.92	−7.4 (23.8–38.6)	<0.001	51.3 ± 10.1	48.0 ± 7.8	0.87	−3.3 (6.9––13.4)	<0.001
Li, Y. et al. [38]	195	3D-STE	140.9 ± 76.9	134.4 ± 68.3	0.94	−6.4 {51.2 (−57.6, 44.8)}	<0.001	102.6 ± 76.2	92.0 ± 60.7	0.96	−10.6 {50.3 (−60.9, 39.7)}	<0.001	32.4 ± 15.5	35.5 ± 13.1	0.91	3.1 {12.6 (−9.5, 15.7)}	<0.001
De Potter, T. et al. [39]	36 + 30	Multi beat 3DE	144.3 *±* 43.0	91.1 *±* 24.4	0.65	−53 ± 32.8	<0.0001	60.4 *±* 21.2	48.1 *±* 16.4	0.53	−12.3 ± 18.7	0.003	58.2 ± 5.4	47.5 ± 7.4	0.1	−10.7 ± 8.7	<0.001
Laser, K.T. et al. [40]	60 (20 + 40CHD)	CMR (KBR) vs. RT3DE	134.4 ± 73.3	127.5 ± 58.0	0.98	2.7 ± 9.5%	–	63.0 ± 48.4	58.0 ± 33.1	0.97	2.2 ± 13.7%	–	55.4 ± 9.4	55.6 ± 8.5	0.82	0.1 ± 9.5%	–
		CMR (MOD) vs. RT3DE	131.9 ± 68.7	127.5 ± 58.0	0.99	1.1 ± 7.4%	–	61.0 ± 45.4	58.0 ± 33.1	0.97	−1.5 ± 13.3%	–	56.1 ± 10.7	55.6 ± 8.5	0.87	0.8 ± 9.2%	–
van der Zwaan, H.B. et al. [41]	62	RT 3DE	219 ± 86	185 ± 71	0.93	34 (−32–99.0)	<0.001	114 ± 62	103 ± 48	0.91	11 (−43–66)	<0.001	49 ± 10	46 ± 08	0.74	4 (−10–17)	<0.001
Otani, K. et al. [43]	100	Fully Auto 3DE	105 (88–132)	93 (75–113)	0.82	−12.6	<0.001	57 (45–83)	51 (39–72)	0.82	–7.5	<0.001	43.4 (35.8–51.5)	44.1 (34.2–49.4)	0.72	−0.3	1.00
		Manual	105 (88–132)	102 (84–121)	0.83	–2.9	0.77	57 (45–83)	56 (44–72)	0.87	–2.4	1.00	43.4 (35.8–51.5)	45.6 (36.1–51.6)	0.87	0.8	0.79
Knight, D.S. et al. [46]	100	3DE Single beat	–	–	0.97	−2.3 ± 27.4	<0.0001	–	–	0.98	5.2 ± 19.5	<0.0001	–	–	0.91	−4.6 ± 13.8	<0.0001
Namisaki, H. et al. [48]	174	Fully Automated 3D (RVFV-Auto)	103 (87–130)	93 (74–120)	–	–	<0.001	56 (45–83)	53 (39–72)	–	–	–	43 (36–51)	43 (34–49)	–	–	<0.001
		(RVFV-Manual edit)	103 (87–130)	105 (85–135)	–	–	<0.005	56 (45–83)	57 (44–78)	–	–	–	43 (36–51)	45 (36–51)	–	–	–
		(4CV-Automated)	103 (87–130)	93 (70–120)	–	–	<0.001	56 (45–83)	53 (390–74)	–	–	–	43 (36–51)	42 (34–48)	–	–	–
		(4CV-Manual edit)	103 (87–130)	103 (82–132)	–	–	<0.001	56 (45–83)	58 (42–82)	–	–	–	43 (36–51)	44 (37–50)	–	–	0.001
Van der Zwaan, H.B. et al. [49]	41 100	RT 3DE (Control)	158 ± 32	127 ± 32	–	34 ± 65	<0.001	65 ± 18	58 ± 16	–	11 ± 55	<0.05	60 ± 6	55 ± 5	–	4 ± 13%	<0.001
		Case (CHD)	193 ± 72	170 ± 21	–	–	<0.001	94 ± 47	96 ± 44	–	–	<0.001	53 ± 9	48 ± 9	–	–	<0.001
Ahmad, A. et al. [52]	170	3DE auto RV	119.8 (91.1–175.8)	112.9 (84.6–150.0)	0.79	–17.8 (−112.6–77.0)	<0.0001	78.1 (51.7–147.7)	64.7 (42.9–110.3)	0.85	−23.6 (−117.2–70.0)	<0.0001	34.0 (17.5–44.5)	38.9 (27.6–50.1)	0.78	6.8 (−12.4–26.0)	<0.0001
		Manual Edit	119.8 (91.1–175.8)	116.9 (88.6–148.9)	0.92	−12 (−79.1–54.5)	<0.0001	78.1 (51.7–147.7)	73.6 (48.1–113.7)	0.95	−13.8 (−73.7–46.1)	<0.0001	34.0 (17.5–44.5)	35.6 (22.9–45.6)	0.94	2.6 (−7.6–12.8)	<0.0001
Trzebiatowska-K, A. et al. [56]	36	3DE	197 ± 59	188 ± 53	0.82	8.46 (−55.8–72.7)	<0.001	114 ± 41	100 ± 30	0.72	13.2 ± 29	<0.001	43 ± 8	46 ± 8	–	−3.29 (−19.7–13.1)	–

## Data Availability

Not applicable.

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
