# Peer review of "Three-Dimensional Echocardiography Assessment of Right Ventricular Volumes and Function: Technological Perspective and Clinical Application"

_diagnostics, 2022, doi:10.3390/diagnostics12040806_

Round 1

Reviewer 1 Report

I want to thank the handling editor for providing me the opportunity to review this informative paper. I would also like to congratulate the authors for their comprehensive review.

This is a review on three-dimensional echocardiography in the assessment of right ventricle volume and function. The authors conclude that this technique can contribute to faster and more accurate diagnosis of right ventricle pathology, and cardiac pathology in general. The authors covering its most significant aspects. Furthermore, the conclusions drawn are a logical sequence of the content and help identify areas for future application, and relevant literature seems to be cited.

Suggestions is revision of Title because there is abbreviation of right ventricle, and of Abstract because there is redundancy of text, which mostly include other imaging technique.

Unfortunately, the text is full of numerous errors in the spaces between words, abbreviations, citations, cited authors and style of references. Some examples are:

  1. line 63: 2DE(20–22).
  2. line 87: cristasupraventricularis
  3. line 137: view and 4CV was
  4. line 148: software (13,19,43–48,20,27,36–38,40–42)
  5. line 437: K.J. L, J.X.C. C, K. P, L.G. K, D. D, G. S, et

These need editing revisions by authors, as is explained in Guidelines for Authors.

Author Response

Response to Reviewer 1 Comments

Comments and Suggestions for Authors

I want to thank the handling editor for providing me the opportunity to review this informative paper. I would also like to congratulate the authors for their comprehensive review.

This is a review on three-dimensional echocardiography in the assessment of right ventricle volume and function. The authors conclude that this technique can contribute to faster and more accurate diagnosis of right ventricle pathology, and cardiac pathology in general. The authors covering its most significant aspects. Furthermore, the conclusions drawn are a logical sequence of the content and help identify areas for future application, and relevant literature seems to be cited.

Response 1: Thank you very much for your appreciation and gives us the opportunity to revise and resubmit our manuscript to the diagnostics. We greatly appreciate your constructive comments and suggestions, which we believe will help improve the quality of our manuscript. We have carefully considered the comments and have revised the manuscript to address the comments and suggestions. We have prepared point-by-point responses to these comments and have highlighted the changes in the revised manuscript.

Suggestions is revision of Title because there is abbreviation of right ventricle, and of Abstract because there is redundancy of text, which mostly include other imaging technique.

Response: We really appreciate your this constructive suggestion. We have revised the Title and also abstract to minimize the redundancy. Please check in the revised manuscript. Once again thank you for your suggestion.

Unfortunately, the text is full of numerous errors in the spaces between words, abbreviations, citations, cited authors and style of references. Some examples are:

  1. line 63: 2DE(20–22).
  2. line 87: cristasupraventricularis
  3. line 137: view and 4CV was
  4. line 148: software (13,19,43–48,20,27,36–38,40–42)
  5. line 437: K.J. L, J.X.C. C, K. P, L.G. K, D. D, G. S, et

These need editing revisions by authors, as is explained in Guidelines for Authors.

Response: We are very thankful to you for highlighting the errors which will help us in improving the quality of our abstract. We have carefully revised the manuscript point-by-point and eliminated the errors including the above mention errors. Once again thank you for your comments and suggestions.

Reviewer 2 Report

GENERAL COMMENTS

This Review covers the role of 3DTTE in the evaluation of the right ventricle (RV) . The topic is important and in the last decade 3DTTE of the RV became an alternative method to CMR for obvious practical reasons and for the diagnostic and prognostic impact of this “simple” TTE approach.

The Review is complete, interesting and well written, however some parts of the text can be improved.

SPECIFIC COMMENTS

  1. A figure (or Figures) including the method of acquisition and final morphologic result of 3DRV reconstruction may be useful for the readers. Main measurments obtianed form the method may be included in this figure.
  2. Methods: Acquisition (depending on different ultrasound unit and different softwares) may be better described: on – line vs off line reconstruction, single beat and multi beats; artificial intelligence new methods; new softwares not only allow semi-automated or automated methods for RV volumes and function but also derivate all 2D and Doppler  (TAPSE, RV Area changes, RV strain) from the 3D dataset. Artifial intelligence method has also to be more detialed.
  3. References: some references concerning feasibility, and clinical proposals of the method may be quoted. (sorry the majority of these new references are from the same Group that largely contributed to this topic and anticipated other papers, but has not been mentioned)

Pag 3  Reference Values of 3DTTE  

This article  article first reported Reference values of 3DTTE and may be inserted.

Tamborini G, Ajmone Marsan, N, Gripari P, Maffessanti F, Brusoni D, Muratori M, Caiani E, Fiorentini C,  Pepi M: Reference values for right ventricular volumes and ejection fraction with real-time three-dimensional echocardiography: evaluation in a large series of normal subjects. J Am Soc Echocardiogr 2010; 23: 109-115.

Pag 4 Feasibility

Tamborini G, Brusoni D, Torres Molina J, Galli C, Maltagliati A, Muratori M, Susini F, Colombo C, Maffessanti F, Pepi M: Feasibility of a new generation three-dimensional echocagraphy for right ventricualr volumetric and functional measurements. Am J Cardiol 2008; 102: 499-505.

Tamborini G, Cefalù C, Celeste F, Fusini L, Garlaschè A, Muratori M, Ghulam Ali S, Gripari P, Berna G, Pepi M. : Multi-parametric "on board" evaluation of right ventricular function using three-dimensional echocardiography: feasibility and comparison to traditional two-and three dimensional echocardiographic measurements. Int J Cardiovasc Imaging. 2018 Nov 15. doi: 10.1007/s10554-018-1496-9. [Epub ahead of print]

Pag 4  Multicenter values and Age, Body size and Gender References

Francesco Maffessanti1*, Denisa Muraru2, Roberta Esposito3, Paola Gripari1, Davide Ermacora2, Ciro Santoro3, Gloria Tamborini1, Maurizio Galderisi3, Mauro Pepi1 and Luigi P. Badano2: Age-, Body Size- and Gender-specific Reference Values for Right Ventricular Volumes and Ejection Fraction by Three-dimensional Echocardiography: A Multicenter Echocardiographic Study in 507 Healthy Volunteers . Circ Imaging 2013.

Clinical Application

Since one main application is to ovecome the limitaiton of TAPSE after cardiac surgery the importance of 3D TTE of  RV analysis after cardiac surgery may be inserted in a short paragraph.

Tamborini G, Muratori M, Brusoni D, Celeste F, Maffessanti F, Caiani E, Alamanni F, Pepi M: Is right ventricular sistolic function reduced after cardiac surgery? A two- and theree-dimensional echocardiographic study. Eur J Echocardiogr 2009; 10: 630-634.

Comparison vs CMR

Gopal AS, Chukwu EO, Iwuchukwu CJ, Katz AS, Toole RS, Shapiro W,
et al. Normal values of right ventricular size and function by real time
3-dimensional echocardiography: comparison with cardiac resonance
imaging. J Am Soc Echocardiogr 2007;20:445-55.

Author Response

Response to Reviewer 2 Comments

GENERAL COMMENTS

This Review covers the role of 3DTTE in the evaluation of the right ventricle (RV) . The topic is important and in the last decade 3DTTE of the RV became an alternative method to CMR for obvious practical reasons and for the diagnostic and prognostic impact of this “simple” TTE approach.

The Review is complete, interesting and well written, however some parts of the text can be improved.

Response 1: Thank you very much for giving us the opportunity to revise and resubmit our manuscript to the diagnostics. We greatly appreciate your constructive comments and suggestions, which we believe will help improve the quality of our manuscript. We have carefully considered the comments and have revised the manuscript to address the comments and suggestions. We have prepared point-by-point responses to these comments and have highlighted the changes in the revised manuscript.

SPECIFIC COMMENTS

  1. A figure (or Figures) including the method of acquisition and final morphologic result of 3DRV reconstruction may be useful for the readers. Main measurments obtianed form the method may be included in this figure.

Response 1: we are very thankful for your this constructive comments. We have included figure in the acquisition section explaining the 3DE RV reconstruction and analysis. Please refer to Figure 1.

  1. Methods: Acquisition (depending on different ultrasound unit and different softwares) may be better described: on – line vs off line reconstruction, single beat and multi beats; artificial intelligence new methods; new softwares not only allow semi-automated or automated methods for RV volumes and function but also derivate all 2D and Doppler  (TAPSE, RV Area changes, RV strain) from the 3D dataset. The artifial intelligence method has also to be more detailed.

Response 2: We appreciate this valuable comment. We have described the different ultrasound units and 3DE softwares commercially available for RV analysis followed by on board and off line reconstruction, single beat and multi beats and summarized other parameters (TAPSE, FAC) assessed with the help of 3DE. Please refer to “Acquisition of RV data set for 3DE image” section.

  1. References: some references concerning feasibility, and clinical proposals of the method may be quoted. (sorry the majority of these new references are from the same Group that largely contributed to this topic and anticipated other papers, but has not been mentioned).

Response 3: Thank you for your suggestion. We have included all the provided references in the manuscript please check it in the revised version.

Pag 3  Reference Values of 3DTTE  

This article  article first reported Reference values of 3DTTE and may be inserted.

Tamborini G, Ajmone Marsan, N, Gripari P, Maffessanti F, Brusoni D, Muratori M, Caiani E, Fiorentini C,  Pepi M: Reference values for right ventricular volumes and ejection fraction with real-time three-dimensional echocardiography: evaluation in a large series of normal subjects. J Am Soc Echocardiogr 2010; 23: 109-115.

Response: Thankfully we have cited the reference in the manuscript please refer to introduction section reference No 23.

Pag 4 Feasibility

Tamborini G, Brusoni D, Torres Molina J, Galli C, Maltagliati A, Muratori M, Susini F, Colombo C, Maffessanti F, Pepi M: Feasibility of a new generation three-dimensional echocagraphy for right ventricualr volumetric and functional measurements. Am J Cardiol 2008; 102: 499-505.

Tamborini G, Cefalù C, Celeste F, Fusini L, Garlaschè A, Muratori M, Ghulam Ali S, Gripari P, Berna G, Pepi M. : Multi-parametric "on board" evaluation of right ventricular function using three-dimensional echocardiography: feasibility and comparison to traditional two-and three dimensional echocardiographic measurements. Int J Cardiovasc Imaging. 2018 Nov 15. doi: 10.1007/s10554-018-1496-9. [Epub ahead of print]

Response: Thank you for providing the suggested references we have included these in our review in feasibility section. Please refer to reference no 46 and 56.

Pag 4  Multicenter values and Age, Body size and Gender References

Francesco Maffessanti1*, Denisa Muraru2, Roberta Esposito3, Paola Gripari1, Davide Ermacora2, Ciro Santoro3, Gloria Tamborini1, Maurizio Galderisi3, Mauro Pepi1 and Luigi P. Badano2: Age-, Body Size- and Gender-specific Reference Values for Right Ventricular Volumes and Ejection Fraction by Three-dimensional Echocardiography: A Multicenter Echocardiographic Study in 507 Healthy Volunteers . Circ Imaging 2013.

Response: Thankfully we have included this valuable informative reference in our review. Please refer to reference no 54 in the feasibility section.

Clinical Application

Since one main application is to ovecome the limitaiton of TAPSE after cardiac surgery the importance of 3D TTE of  RV analysis after cardiac surgery may be inserted in a short paragraph.

Response: We are very thankful to you for your this kind of constructive comment for the improvement of review quality, we have included a short paragraph on the role of 3DE post-cardiac surgery. Please refer to the last paragraph.

Tamborini G, Muratori M, Brusoni D, Celeste F, Maffessanti F, Caiani E, Alamanni F, Pepi M: Is right ventricular sistolic function reduced after cardiac surgery? A two- and theree-dimensional echocardiographic study. Eur J Echocardiogr 2009; 10: 630-634.

Response: We have cited this reference in the post-cardiac surgery section please refer to reference no 112.

Comparison vs CMR

Gopal AS, Chukwu EO, Iwuchukwu CJ, Katz AS, Toole RS, Shapiro W,
et al. Normal values of right ventricular size and function by real time
3-dimensional echocardiography: comparison with cardiac resonance
imaging. J Am Soc Echocardiogr 2007;20:445-55.

Response: Thankfully we have cited this valuable informative reference in our study, please refer to reference no 55.

Round 2

Reviewer 2 Report

The new version of the Review markedly improved . The Figure is appropriate and very nice.